# Ageing in place or stuck in place: Preferred care setting for community-dwelling older persons in a low-resource country in Sub Saharan Africa

**Eniola Olubukola Cadmus** [1,2] *, **Lawrence Adekunle Adebusoye** [1,2], **Eme Theodora Owoaje** [1]

1 Department of Community Medicine, College of Medicine, University of Ibadan, Ibadan, Nigeria, 2 Chief Tony Anenih Geriatric Centre, University College Hospital, Ibadan, Nigeria

* eniyolacadmus@gmail.com

**Data Availability Statement:** All data files for the qualitative and quantitative aspects of the study are available from the Figshare database (accession

## Abstract

### Background

Ageing in Place is the emerging social policy drive for long-term care coordination of older persons globally. This decision may be the only viable option in many low- and middle-income countries like Nigeria. Nevertheless, the risk of older persons being 'stuck in place' is high if their preferences are not considered or other alternatives are neither acceptable nor available. This study determined factors associated with the preferred care setting among community-dwelling older persons and explored their views about their choices.

### Methods

The study utilised a mixed-methods approach. Participants were older persons ($\geq$ 60 years) in a selected rural and urban community in Oyo State, south-western Nigeria. Quantitative data were collected using an interviewer-administered, semi-structured questionnaire and analysed using Stata version 14 at p<0.05. Qualitative data collection involved 22 Focus Group Discussions (FGD). The discussions were audiotaped, transcribed verbatim and analysed thematically using ATLAS.ti version 8. Selected quotations were used to illustrate themes.

### Results

1,180 participants (588 rural vs 592 urban) were interviewed with a mean age of 73.2 ±9.3 years. More rural participants preferred to AIP (61.6%) compared to urban participants (39.2%), p = 0.001. Factors associated with the decision for rural participants were older age [OR:2.07 (95%CI:1.37–3.14)], being male [OR:2.41(95%CI:1.53–3.81)] and having assistance at home [OR:1.79 (95%CI:1.15–2.79)]. In comparison, significant factors for urban participants were older age ($\geq$70years) [OR:1.54(95%CI:1.03–2.31)] and home-ownership [OR:5.83 (95%CI:3.82–8.91)]. The FGD revealed that the traditional expectation of reciprocity of care mostly influenced the desire to AIP. Advantages include improved social

number: https://doi.org/10.6084/m9.figshare.22339795.v1).

**Funding:** This research was supported by the Consortium for Advanced Research Training in Africa (CARTA). CARTA is jointly led by the African Population and Health Research Center and the University of the Witwatersrand and funded by the Carnegie Corporation of New York (Grant No. G-19-57145), Sida (Grant No:54100113), Uppsala Monitoring Center, Norwegian Agency for Development Cooperation (Norad), and by the Wellcome Trust [reference no. 107768/Z/15/Z] and the UK Foreign, Commonwealth & Development Office, with support from the Developing Excellence in Leadership, Training and Science in Africa (DELTAS Africa) programme. The statements made and views expressed are solely the responsibility of the Fellow. The funders had no role in study design, data collection and analysis, decision to publish, or preparation of the manuscript.

**Competing interests:** The authors have declared that no competing interests exist.

connectedness, quality of care, community participation and reduced isolation. Interestingly, participants were not opposed to the option of institutional care.

## Conclusion

Ageing in place is preferred and influenced by advanced age and home ownership in our setting. Information provided could guide age-friendly housing policies and community-based programmes for the care of older persons.

## Introduction

The phenomenon of population ageing is occurring in both high-income countries (HIC) as well as low and middle-income countries (LMIC) [1]. By 2050, estimates suggest that older persons will constitute 21.5% of the global population, doubling from 12.3% in 2015. In the near future, four out of every five older people globally will reside in LMIC [2]. By 2030, older persons in the African region will rise to 105 million and about 220 million by 2050 [2]. On the continent, East Africa ranks first among the regions with the highest population of older persons (18.9 million), followed by North Africa (17.9 million) and West Africa (16 million) [1].

Nigeria has the highest African population and ranks seventh globally [2]. In Nigeria, older persons aged 60 years and above constitute about 5.0% of its over 200 million population [2]. Older persons were estimated to number about 8.2 million in 2015 and will rise to 12 million by 2030. This number will increase further and reach about 25.2 million by 2050 [2].

The ongoing demographic transition will likely result in an increased prevalence of chronic diseases, functional decline, and increased reliance of older persons on others for their long term care (LTC) needs [3]. According to the World Health Organization (WHO), LTC is 'the series of activities undertaken by others to ensure that people at risk of significant loss or with ongoing loss of intrinsic capacity can maintain a level of functionality consistent with their basic rights, fundamental freedom and human dignity' [4]. In its meeting in 2016, the World Health Assembly resolved that all countries must make provisions for LTC for their older population. Although the 194 member states agreed with the recommendation, few have made provisions and measures to support the older population segment.

Long-term care provisions may be provided formally or informally within the home, community, or dedicated institutions [5, 6]. Common LTC settings include senior centres, assisted living facilities (ALF), and in-house services, referred to as 'Ageing in Place'. Senior centres provide day-care and community interaction services in a non-residential setting. The ALF offer older persons needed support but not strictly in the health facility. Furthermore, these facilities provide individualised services to suit the preferences and needs of the client. Services provided include support, social, recreational and health-related services [7].

Ageing in place is defined as: 'living at home or in the community with some level of independence, privacy, safety and control over one's environment' [8]. The concept aims to support older persons in their preferred accommodation and communities where they reside for as long as possible despite actual or potential changes in health and functioning in old age [9]. According to Rosenwohl and colleagues, the process of ageing in place may be explained by three important constructs, namely identity, connectedness, and sense of place. In addition, the experience of ageing is a balance between threats and agency. On the one hand, agency is the ability of older persons to make decisions and the capacity to respond to threats. Also, agency assists in maintaining important elements of their daily lives while ageing in place [10].

As such, agency depends on factors such as financial stability, social connections and available services. On the other hand, threats include changes in health functions and needing assistance with activities of daily living [10].

Ageing in place as a model is increasingly gaining recognition in many HIC, especially with the increased burden of chronic disease and disability due to longevity [11–13]. The decision to AIP benefits older persons' health and overall quality of life (QoL) [14, 15]. Furthermore, accepting the approach in many HIC is based on its potential for reduced cost of care compared with institutionalised care [16, 17]. Existing research, mainly from HIC, shows that older persons prefer to AIP interminably regardless of their functional capacity [18].

Several socio-economic and demographic factors are documented to stimulate the expectations and decisions of an older person regarding the setting for LTC. A study conducted among older persons in Detroit, USA, established that living alone, home-ownership, and independent transportation mode through driving influenced expectations to AIP [19]. Likewise, Golant identified other factors, including the environmental characteristics and the availability of dedicated caregivers to provide appropriate care [20]. Additionally, Shiamberg and McKinney [21], documented factors that influenced the older person's decision to AIP, including individual factors such as financial capacity, socio-economic characteristics, and healthcare needs [21]. Other factors that influenced decision-making in the study were contextual and included safety, economic independence, and availability of jobs and health care [21].

Another factor reported to influence the preferred setting for the care of older individuals is the availability of informal support. For instance, in a qualitative study among older persons in seven naturally occurring retirement communities, Greenfield noted that informal support from neighbours contributed to promoting ageing in place [22]. Likewise, Gardner reported that relationships with neighbours positively influenced the decision to AIP [23]. Furthermore, Ball and colleagues conducted a qualitative study investigating ALF [24]. The authors demonstrated that managing functional decline was the key to ageing in such facilities. Therefore, no matter where the older person is ageing, either in the home community or in a specially designated facility, provisions must be put in place to manage functional decline [24].

Ageing in place has been shown to include benefits to the older person, family, communities, and government. Benefits to the individual include health and well-being, promoting attachment and familiarity with the environment [25]. Research shows that the inability of older persons to remain in a familiar environment leads to loneliness, depression, difficulty adjusting and functional deterioration [26]. Furthermore, ageing in place has economic benefits as home care is far cheaper than residential care [11]. Other benefits of ageing in place include social inclusion, social connection, autonomy, and companionship [27].

However, the picture in many LMICs remains largely unchartered as few studies have established an accurate picture. Nevertheless, the ability to AIP seems to exist and is purported to be the preferred choice among individuals in low-income settings. However, there may be evidence that ageing in place in such environments may be more an arrangement of convenience rather than choice. For instance, in their study among low-income Chinese city dwellers, Lum and colleagues set out to establish whether individuals were 'ageing in place' or 'stuck in place' when the capacity to relocate was unavailable [28].

Furthermore, Golant pointed out the flaws of studies that suggest that older persons desire to AIP and criticised the failure of such studies to distinguish preferences among older persons with varying health needs and disabilities [20]. Similarly, Lehning, Smith, and Dunkle examined the association between age-friendly community characteristics and the desire to AIP [19]. Although the authors found no differences across income levels, it was unclear whether the increased expectations to AIP were an actual choice among low-income earners or a lack of any other affordable alternative.

Most research carried out to date on preferred care settings for older persons, including ageing in place has been conducted in HIC [13, 29–31]. The studies show that older persons prefer to AIP despite their functional capacity. There is a narrow focus on LMIC, such as in Nigeria, particularly emphasising the preference of older persons. Rural and urban communities in these areas have differences in structure, services, and amenities. Considering the poor policy and support systems for older persons in LMIC, there are concerns that older persons may choose to AIP not because they choose to but because they must. Also, vulnerable adults such as the poor, more senior, poorly educated, unmarried and females may not have any option and may be trapped.

Recent estimates suggest that many older persons in SSA require LTC at different times across their life span [4]. This need is bolstered by the WHO Global Strategy and Action Plan, which calls for all countries to face the challenges of LTC provisions for their ageing population [32]. Likewise, one of the action areas for the United Nations (UN) Decade of Healthy Ageing (2021–2030) is the provision of LTC for older persons [33]. This call for multidisciplinary efforts to achieve healthy ageing necessitates an enquiry into the preferences of the most critical stakeholders (older persons). This study compared, using quantitative measures, the preferred care setting for ageing and factors associated with the decision among rural and urban community-dwelling older persons aged 60 years and above in Oyo State, Nigeria. Also, the study explored the views about their choices using qualitative measures.

## Materials and methods

This paper describes part of a more extensive study which compared the preferred setting for care, quality of life, attitude to ageing and disability among older persons in a rural and an urban community in Oyo State, south-western Nigeria. The first study highlighted the rural–urban differences in quality of life and associated factors among community-dwelling older persons in Oyo state, South-Western Nigeria [34]. The second study described the populations attitude towards ageing and perceived health status [35].

The study presented in this paper used a mixed research method which has been described previously [34, 35]. A mixed-methods approach was employed because of the exploratory nature of this study. The quantitative measures were to substantiate the preferred settings for care by the participants and associated factors. The qualitative aspect of the study explored the views regarding their preferred care setting. Triangulating quantitative and qualitative data collection techniques enabled a richer and deeper understanding of the research findings [36].

### Study area

Nigeria is in West Africa and has an estimated population of 201 million people [2], making it the seventh most populous country globally. Nigeria is divided into 36 administrative divisions (States) and the Federal Capital Territory. The urban population constitutes 51.6% of the country's total population [2]. The percentage population aged 65 years and above is 2.7%.

The study was conducted in Oyo State, located in the south-western zone of Nigeria. Oyo State has a landmass of 27,249 square kilometres and is divided into three Senatorial Districts: Oyo North, Oyo Central and Oyo South, consisting of 33 Local Government Areas (LGA). There are 12 urban, nine semi-urban and 12 rural LGAs in the State. Oyo State has a projected population of 7,840,900. This estimate was based on a 3% annual growth rate applied to the 2006 census figures [37]. The proportion of older persons in Oyo state is about 6% [37].

### Study sites

The study was conducted in one urban (Ibadan North) and one rural (Ibarapa Central) LGA in Oyo State. The residents are mainly of Yoruba ethnicity; they are similar culturally and

speak the same language, Yoruba. These study sites are under the purview of the Department of Community Medicine, College of Medicine, University of Ibadan and have been geographically mapped and listed. Utilising these sites for the study offers a unique opportunity for subsequent development and testing of interventions to suit the peculiarities of older persons in the community.

**Ibadan North LGA.** Ibadan North LGA (IBNLGA) is a densely populated urban area with a population of 432,900. About 4% (18,524) of the people of Ibadan North LGA are aged 60 years and above. The residents are predominantly of Yoruba ethnicity, but other resident ethnic groups include Igbo and Hausa. Ward 3 was purposively selected for the study out of the 12 wards in the LGA. Ward 3 comprises the settlements of Yemetu Aladorin, Adeoyo and part of Oje. In 2013, a demographic survey of the residents in Ward 3 of IBNLGA was conducted by the Department of Community Medicine, UCH, in collaboration with the Department of Geography, University of Ibadan, and staff of the Oyo State office of the National Population Council. Based on the figures obtained in 2013, the projected population of Ward 3 at the time of the study was 30, 861.

**Ibarapa Central Local Government Area.** The rural setting chosen for the study is Ibarapa Central LGA which has its headquarters in Igboora. Recent estimates suggest the LGA has about 140,900 people, of which about 4% (5,479) are aged 60 years and above. The LGA is divided politically into ten wards. Igboora comprises Wards 4–10 of the LGA. In 2006 the population of the town was approximately 72,207 people. In 2017 the population was estimated to be around 178,514 [38, 39].

**Study design and participants.** The study participants comprised older males and females aged 60 years and above, at their last birthday, in selected households and study sites.

**Inclusion and exclusion criteria.** Adults aged 60 years and above and permanently living continuously in the selected communities for at least 12 months before the study were eligible to participate. The eligibility was based on the criteria used in a similar community-based survey conducted by Gomez-Olive et al. (2018) in South Africa [40]. However, older persons who had any form of incapacitation or health conditions such as dementia, speech or hearing disorders and other communication problems and could not personally provide information were excluded from the study. Proxies were not utilised for this study because the objective was to document the older person's views about their preferences and ageing experience, which a proxy may not accurately capture.

## Data collection and analysis

**Quantitative data.** *Sample size calculation*. The sample size for the quantitative study was determined using the sample size calculator in Stata Version 12 and the formula for comparing two independent proportions assuming a difference of 10%, power of 90%, and α of 5% [41, 42]. The minimum calculated sample size was 522. However, in total, 1,180 respondents (592 urban and 588 rural) were selected for the survey.

*Sampling technique*. A comparative cross-sectional survey was conducted in the purposively selected LGA. A multi-stage sampling procedure was utilised for the participant selection to select the LGA, enumeration areas (EA), ward and households. The purposive selection of the LGA was based on characteristics earlier highlighted (previous mapping and listing). In the LGA selected, the EA chosen included the ward of interest. The ward was also chosen because it has been listed and divided using a standard approach into areas. In each selected ward, four areas were selected through balloting without replacement. The available database from the Department of Community Medicine census was probed to identify households with the desired target population. The selected households were then visited. If an intended

respondent was unavailable due to relocation or demise, the field workers were trained to move to the next listed household. Similar to previous studies, if a selected respondent was not met at home, the house would be visited thrice, after which the field workers would move on to the next house on the list [43, 44]. Also, if more than one eligible participant was in a particular household, balloting was carried out to select a participant.

*Data collection*. Quantitative data were collected over two months in July 2018 using a semi-structured questionnaire. Trained research assistants (RA) administered the questionnaires after informed consent had been obtained. The survey was deployed using the Research Electronic Data Capturing (REDCap) on mobile devices (tablets).

A total of 10 RA with a minimum educational attainment of at least a secondary school leaving certificate were trained in the proper approach to administering the questionnaires. The training was done over three days before a pre-test and subsequent commencement of fieldwork. Also, two community inhabitants domiciled in the respective LGA were recruited to guide the research team on the field. The field guides assisted with community entry and identification of selected households. The RA visited each selected household, and face-to-face interviews were conducted.

The developed instrument reflected the study objectives and consisted of seven sections including socio-demographic characteristics, family dynamics, current health status and selected health risk behaviours. Also, the instrument assessed the participant's attitude to ageing, quality of life, self-reported functioning, and disability. Lastly, the preferred care setting was determined. Socio-demographic variables obtained included (age, sex, employment status, marital status, living arrangement, and socio-economic status). Respondents' economic status was determined using the equity tool index based on the wealth indices generated from Demographic and Health Surveys (DHS) and other similar surveys. The surveys include questions about the respondent's household characteristics and owned assets [45]. Wealth status was categorised into quintiles by relating each respondent's total possessions to the median number of possessions of the sample. Participants' attitude to ageing was assessed using the attitude to ageing questionnaire (AAQ) [46]. The instrument measured the attitude to ageing in psychosocial, physical change and psychological domains.

Participants were asked to rate their quality of life on a five-point Likert scale ranging from very satisfied to very bad. The responses were categorised as having good QoL (if their response to the question of Qol were "good" or "very good"). Likewise, respondents were said to have poor QoL if they reported these to be "bad", "very bad", or "neither bad nor good". Participants were also asked to rate their health on a five-point Likert scale ranging from very good to very bad. The responses were categorised as having good self-rated health (SRH) (if their response were "good" or "very good". Likewise, respondents were said to have poor SRH if they reported their health as "bad", "very bad", or "neither bad nor good". The respondent's level of functioning was measured using the 12-item World Health Organization Disability Assessment Scale (WHODAS), Version 2.0 [47]. The domains evaluated the level of disability among respondents in six domains. Other variables measured include living dynamics such as living arrangements, having assistance and home-ownership. Ownership of housing was categorised as owning the homes in which they reside or not. Participants were also asked if they had assistance at home, to which the response was Yes or No.

The research instrument was translated into the local Yoruba language for easy communication. It was back translated to English to ensure the original meanings were maintained before administration. A pre-test was conducted among the older persons in a different LGA from those selected for the study. Following the pre-test, ambiguous questions were rephrased or removed where necessary. Each of the interviews lasted between 30 minutes to 45 minutes.

*Data analysis*. The primary outcome variable for the study is the participants preferred care setting. The options included staying within their homes or community (AIP). Quantitative data were analysed using Stata version 14 [42] and presented using frequency tables and charts. Summarisation was done using proportion and means. Associations were measured between independent and dependent variables using the Student t-test, Chi-square test and logistic regression. Variables significant at 10% on bivariate analysis were included in the regression model and fitted for the rural, urban, and total population at a 5% level of statistical significance.

*Qualitative data collection and analysis*. The qualitative aspect of the study included the conduct of Focus Group Discussions (FGD). The FGD explored in depth the participants' experiences and views on preferred care setting in their respective communities across the spectrum of functionality. Also, the dynamics surrounding their choices were investigated. Each focus group comprised eight to ten people (60 years and above) and the research team. The research team comprised of a moderator/ facilitator, an observer, and a recorder. Efforts were made to ensure that the groups were as homogenous as possible regarding socio-demographic characteristics such as gender (male, female), age stratification (60–69 years, 70 years and above) and location (rural, urban). The FGD participants were from the selected communities but did not necessarily partake in the questionnaire survey.

Participants were recruited purposively through the help of field scouts and relevant gatekeepers in the community, like the Chairman of the Ward Development Committee in the urban area and a notable and well-respected pastor in the rural community. The discussions in the urban area were held in a centrally located comprehensive primary health care centre. The rural discussions took place in the house of a community leader.

The author and two other assistants (one male and one female) who were experts in qualitative research facilitated the FGD. The three facilitators were bilingual and obtained informed consent from each participant before the discussion. The discussions were facilitated with a semi-structured interview guide with suitable probes to ensure uniformity in questions asked for each group. The sessions were recorded using a tape recorder. A designated note-taker took detailed notes, which aided transcription. The duration of each discussion was between 60 to 90 minutes. A debriefing among the team was held after each FGD to discuss emerging themes. Respondents were provided with refreshments and reimbursed for their travel-related expenses.

Data saturation was attained after 22 FGD had been conducted. The FGD were conducted in Yoruba, transcribed to English by the RA who assisted or did the facilitation and peer-reviewed among the facilitators to confirm translation accuracy. The principal investigator approached a subset of the participants for member checking. The aim was to verify the results obtained as a true reflection of the discussion. The research team also had frequent peer debriefings to ensure that the accounts of the discussion were accurate. Clarifications were sought and discussed extensively, potential biases were probed, and the audit trail was consulted as needed.

The research team included the first author and two research assistants with expertise in qualitative research who served as the note-taker and recorder/ observer. The three members of the team took turns facilitating the group discussions. The facilitator was responsible for the dynamics of the group. Prior to the commencement of the discussion, ground rules were discussed. These included allowing everyone the opportunity to talk one after the other, and there were no right or wrong answers. The facilitator for each discussion ensured compliance.

*Qualitative data analysis*. Data collected were transcribed and analysed using Atlas ti. Version 8 and the thematic framework approach [48]. The transcripts were reviewed several times to familiarise with the content before labelling the key words and phrases. The data analysis

started with line-by-line open coding, breaking segments of the data into corresponding categories, such as preferred care setting, determinants of the choices, benefits of AIP and views about institutional care (Table 1). Apart form ageing in place within the home or community, other options revealed by the FGD included moving in with in-law, first-degree relative/child, church establishment or institutionalised care.

Next, emergent themes that represented the preferred care settings and issues raised based on study objectives were sorted and coded. The themes were categorised and compared before making a decision on the dominant themes to report. In cases of conflicts, a discussion between the lead author and the research ensured a consensus before finalising the various categories for each theme. In the end, the generated themes were analysed. Relevant quotes were identified and used verbatim to buttress the explanation of the findings. Overall,172 older persons participated in the 22 FGD. The participants were mostly Yoruba with a mean age of 70.9 (±2.9) years.

Findings from both quantitative and qualitative data were integrated during the data-interpretation stage. The goal was to obtain different but complementary data that validated the overall results about the participant's preferred care setting and associated factors.

## Ethical considerations

Ethical approvals were obtained from the University of Ibadan/University College Hospital (UI/UCH) Ethical Review Committee (UI/EC/180204). All participants gave written consent for the survey and verbal informed consent for recording the interviews before participating. Participants were assured that their identities would remain anonymous in all study reporting and that their personal information would be kept confidential. All methods in this study followed relevant guidelines and regulations in the Ethical Declarations.

## Results

### Quantitative findings

Overall, 1,180 respondents participated in the survey (592 in the urban area and 588 in the rural are). Participants were mostly female in the rural (406, 69.0%) and urban (417, 70.0%) communities. Rural participants compared to urban participants were older (74.2 ± 9.5 vs 72.3 ± 8.9 years, $p = 0.001$), homeowners (67.0% vs 61.6%, $p = 0.06$) had home assistance (79.8% vs 74.7%, $p = 0.04$) and disability (53.1% vs 50.7%, $p = 0.41$). More participants in the rural community desired to AIP 362 (61.6%) compared with those in the urban setting 232 (39.2%), ($p<0.05$). Among the participants in the rural community, older participants, 277 (69.4%), were more likely to desire to AIP. Likewise, a higher proportion of males, 125 (68.7%), were likely to want to AIP compared with females 237 (58.4%) ($p<0.05$). Also, more unmarried participants preferred to AIP 204 (66.5%) compared with those who were married 158 (56.2%). Likewise, a significantly higher proportion of participants who did not have formal education, 309 (64.9%), were more likely to desire ageing in place compared with those who had formal education 53 (47.3%). Among the participants in the urban community, the desire to AIP was associated with older age ($\geq$70 years) and unemployment (Table 2).

As shown in Table 3, more rural participants who reported having assistance at home desired to AIP 301 (64.2%) compared with those who had no help at home 61 (51.3%), ($p<0.05$). Other factors associated with a desire to AIP among the rural participants were having owned homes, and good self-rated quality of life (SRQoL). Among the participants in the urban community, factors associated with the desire to AIP were living in an owned home and having good SRQoL (p<0.05).

Table 1.  Older persons preferred setting for care and its determinants.

| Theme | Sub-Theme | Relevant Quotes |
|---|---|---|
| **Preferred Care Setting** | | |
| **Ageing in Place** | Home | *'My prayer for my children is for them to build houses wherever they like. . .. I can go to their places to greet them but return to my house the second day. But to now go there to stay there and become a stranger? I will rather be in my house. My prayer is for my children to move forward; they will come and meet me here. They can bring their children for me to take care of them. . .. . . but to lock my door. . .. (Participant shakes his head)'.* (Male FGD_ 70 years and above_ Urban) |
| | Community | *You know you asked me three days ago whether I wanted to leave this place, and I said God would not allow bad things to happen. No matter what, it is one's father's house that you can call your house. Even if you build three or four flats for me, I prefer to come to our 'Agbo-ile' (communal living arrangement)'.* (Male FGD_70 years and above_Urban)_ |
| | Moving in with Family | *". . . if Almighty God didn't send someone out of her husband's house, there is no other place for an older woman to stay than her husband's house till death comes. But if her children build a house and take her to their home, she can stay there too.* (Female FGD_ 60–69 years _Urban) |
| | Intergenerational Arrangements within the family | *'. . .. where an older person lives can be different. For example, the children can take her away with them, and maybe her brother or sister can also tell her to come with them because they might not be able to stay with her where she lives but instead take her with them for proper care'.* (Female FGD_70 years and above_ Rural) |
| **2. Determinants of preferred care setting** | | |
| | **a. Sex** | |
| | i. Female | *'. . . ...she (the older person) will stay in her husband's house and ensure that people are with her. Not all our son's wives will allow their mother-in-law to stay with them. We cannot stay with our female children because she has her family to care for. It is safer to stay in one's husband's house, and the children can employ someone to look after her.* (Female FGD_60–69 years _Urban) |
| | Male | *'. . .exactly as he said, even if my children build a house on the water, I can't forget my father's house. Almost everyone here has children who have built houses in Lagos and want us to follow them. No way! I can't be locked up! '.* (Male FGD _ 70 years and above_ Urban) |
| | **b. Home Ownership** | *'If there is no problem, a woman does not leave the place she had her male and female children till she gets old. It's the house of her children's father that she should age. If there's no problem, God should not make us encounter problems. . .. That is where we ought to age. A woman should age gracefully in her home.* (Female_ FGD 60–69 years_ Rural) |
| | **c. Increased dependence** | *' If there's anyone who has a health challenge as I do, if I have a child who has built a house in Lagos and asks me to come, I will go. The reason is that there is too much stress. I cannot cook by myself, and I cannot fetch water by myself. I cannot do anything by myself. If one has such, who will care for one, why won't I go with him? I will go, as long he is a good child and won't do me wrong but take care of me'. (*Female FGD 70 years and above_Rural) |
| | **Health Challenges** | *'it can change, and the reason is God should allow us to have children around to cater to our needs when we are old and like babies. You might have determined that you will be in your house or ancestral family homes. But when the health declines and the child is in, let's say Ogbomosho(nearby town) as a civil servant, every one of us is praying that our child should move(forward or they are in Abuja(Federal Capital) both male and female, none of them can come and sit with you. Then you have no option other than to be carried by the children. You might now make your children agree to return you to your home for burial if you died in their place'.* (Male FGD_70 years and above Urban) |
| | **d. Traditional expectation of reciprocity of care** | *'An older person who is sick should live with someone who can care for him/her. For instance, if you see an older person who is so sickly, it is your responsibility to find a place to keep her, especially if she is your relative.* We have spent our lives taking care of the children. It is expected they will do the same for us when the time comes'. (Female FGD_ 60–69 years_ Urban) |
| | **e. Societal changes-individualisation/ enucleation of the family** | *'For me, I would not like to live with my son's wife. Do not get me wrong, I do not hate her, but the children don't want to live with their mothers-in-law these days. They just want to live alone with their husband. Anyone who wants to live long should not get involved in something that will later affect her because how she will be treated will be ridiculous. If you have a female child who can care for you when you are old, that would be better'.* (Female FGD 60–69 years _Urban) |

*(Continued)*

**Table 1.** (*Continued*)

| Theme | Sub-Theme | Relevant Quotes |
|---|---|---|
| **Benefits of AIP** | **Quality Care** | |
| | **Increased opportunity for social connectedness and community participation** | '*As we all know, most people will leave their village and settle down in another town, but older people need to stay in a community where they can interact with their age mates. At times they may move back to their villages to stay with family members who will take good care of them because they will assist their community in development based on their experience*'. (Female FGD 70 years and above_ Rural). |
| **Views about Institutionalised care** | | |
| | **Conditions for Disapproval** | |
| | **i. Foreign culture** | '*We can't live there (an institution) o! I can't live there (an institution) because it is not our culture. This arrangement is the white man's culture, and they have accepted it! They know that when they get old, no matter how rich the child is, they will still end up in an old people's home! But the Yoruba believe that our children will take care of us in our old age, and that's why we are giving birth. Somebody will not train his child, send him to school, up to university, and when he's now getting old will now be cared for by the government and the child will now be checking on him occasionally and the child will not do anything. I'm afraid I have to disagree with that arrangement.*' (Male FGD_70years and above_Urban) |
| | **ii. Retribution for poor parenting** | '*. . .my view is that only parents whose children didn't take good care of them go to old age people's homes like the one we have at Idikan (an urban slum). Someone like me, for instance, my children are taking care of me. Why would I go to such a place.*' (Female FGD_60–69 years_Urban) |
| | **iii. Perceived inferiority of care quality** | '*. . .we have said it all, but it is only when we are sick which we don't pray for. In America, institutional care is easy to access, but it is not the same in our country. Even if we go to an old people's home, they will not take good care of us to how our children will take care of us*' (Male FGD_60–69 years_Rural) |
| | **iv. Loss of independence/ sub-optimal quality of care** | '*Just like my colleague said, it's like a bird that used to fly about but is now captured in a cage. The life of that bird is shortened! So, instead of putting older persons in the care of a nurse, they should be with their people. There are things their children can tell them that the nurse would not be able to tell them, and there are some things they can tell their children that they might not be able to tell the nurse. It might even lengthen their days*'. (Female FGD_60–69 years_Urban) |
| | **v. Cost** | '*if they will collect money, I do not support it. I prefer that the individual's son either build a modern house with all the necessary things or have the parent stay with him in his house till death comes*'. (Male FGD_60–69 years_Rural) |
| | **vi. Negative attitude** | '*My opinion is that if my children are taking care of me, why would I go to such a place? All I need is care. I prefer my children's house to older people's home. I don't like it!*' (Female FGD_ 70 years and above_ Urban) |
| | **Reasons for acceptance** | |
| | **i. Ensure provision of basic necessities** | '*That is good, but only if one's child is unavailable and has not built a house to keep his parents. If I discover an institution where I will be catered for, fed, and can enjoy peace, I wouldn't mind living there*'. (Female FGD 70 years and above_ Rural). |
| | **ii. Positive sign of modernisation** | '*. . .. It's a welcome development, we've never travelled to any foreign country, but some of our children that have travelled told us about the centres they have for older people. They take good care of them and even give them a salary. Unlike in our country where you will see unmonitored older people crossing the highway, a car will hit them, and they will die*'. (Female FGD_ 60–69 years_Urban) |
| | **iii. Positive attitude** | '*. . .. . .there is nothing bad in that idea of living in old people's homes so far that no law prohibits our children or family from seeing us at any time, and all the necessary things that old people need to survive are available. Why not? We will definitely stay there . . .. . because everything we cannot afford will be made available. If the old people's home will aid our healthy life, why will we not like it?. . . we will stay there*'. (Male FGD 60–69 years _ Rural) |
| **Preference for provider** | **i. Religious bodies** | '*I know a pastor called XXX with a home for older persons where they take care of them, so it's a welcome development*'. (Female FGD 60–69 years_Urban) |

(*Continued*)

**Table 1.** (Continued)

| Theme | Sub-Theme | Relevant Quotes |
|---|---|---|
| | ii.Government | '. . .. If the government can establish older people's homes, this will relieve the family of some of the burden of caregiving. Also, those working for the government will be relieved and concentrate on their jobs; it is a good idea if the government does not politicise it'. (Male FGD_ 60–69 years_Rural) |
| Reasons for AIP | iii. Communal sense of purpose | '. . .Some residential areas cause discomfort for older adults. Talk of taking them to Bodija or an estate where they can watch television, the problem has started. They would want to go back to their people. If you don't take them back, be ready to spend money on a burial. A standard way of living is good, but they didn't grow up in that environment. No matter what you put in such places, be it air conditioners, you cannot compare with where they rear their goats and get to shout every morning. Their ability to shout out and be free in remote areas keeps them fit better than when they stay in an air-conditioned environment'. (Male FGD_ 60–69 years_Urban) |
| | iv.Preservation of independence | 'A friend of mine, for instance, his child appealed to him to come and live abroad with the family, which he did. He said whenever the son went to work in the morning, he would lock this old man up alone at home. He did not get the joy and pleasure he hoped to get abroad. He was locked in the house till his children returned from work in the evening. He felt imprisoned and confined, which made him decide to return home earlier than planned. Shortly after he returned, he died. He was a very friendly and sociable person, known by most people sitting here, but he died because he did not have people to relate with'. Male FGD_ 70 years and above_ Urban |
| | v.Preservation of routine/ regularity | "My father is about 108 years old now. We've been persuading him to come to Ibadan, but he says he won't last more than a few months if he should come to Ibadan, and we left him on his own. Waking up in the morning, he looks out for his goats and chickens from the window frame. Even though he can't walk, he's satisfied that way. If you try to persuade them (older persons) to relocate to where their children will take good care of them, they (older persons) prefer to stay where they are, even if they would eventually die there. Taking them (older persons) away could shorten their lifespan". (Male FGD_ 60–69 years_Urban). |
| Reasons for relocation | Better facilities/amenities | ". . .. though we have our own house and would love to stay there, I would love to move to a house built by my children because it will be a modern house, so it's my desire that our children build us a house before we leave this earth" (Female FGD 60–69 years _Urban). |

Table 4 shows details of the most significant factors associated with the desire to AIP for rural participants were older age (≥70 years) [OR: 2.07 (95% CI: 1.37–3.14)], being male [OR: 2.41 (95% CI: 1.53–3.81)] having home assistance [OR: 1.79 (95% CI: 1.15–2.79)] and living in an owned home [OR: 2.47 (95% CI: 1.69–3.61)]. However, compared with those who were currently unmarried, participants who were currently married were less likely to desire AIP [OR: 0.61(95% CI: 0.40–0.92). Also, compared with those with low educational attainment, participants who had higher education were less likely to desire to AIP [OR: 0.49 (95% CI: 0.30–0.81)]. However, the most significant factors associated with the desire to AIP for urban participants were older age (≥70 years) [OR: 1.54 (95% CI: 1.03–2.31)] and home-ownership [OR: 5.83 (95% CI: 3.82–8.91)].

Regardless of their desire to AIP or not, participants were asked why they may relocate from their current location. Their responses are shown in Fig 1. The most frequently reported factor that may influence the desire to relocate in the rural community included a desire to be close to family members 211 (35.9%), a need for a manageable size of the housing 187 (31.8%), the need for independence 147 (25.0%) and access to health services 114 (19.4%). The participants less frequently mentioned other reasons such as concerns for safety, house maintenance costs and transport difficulties. Similarly, in the urban area, the most reported reasons to consider relocation include a need for a manageable size of the housing 298 (50.3%), the desire to be close to the family 299 (50.5%) and to facilitate their independence 348 (29.5%).

**Table 2. Selected demographic factors associated with the desire to age in place.**

| Location | Rural = 588 | | | Urban = 592 | | | Total = 1180 | | |
|---|---|---|---|---|---|---|---|---|---|
| Variables | Desired AIP | $\chi^2$ | p-value | Desired AIP | $\chi^2$ | p-value | Desired AIP | $\chi^2$ | p-value |
| **Age Category (years)** | | | | | | | | | |
| <70 years | 85 (45.0) | | | 71 (28.6) | | | 156 (35.7) | | |
| ≥ 70 years | 277 (69.4) | 32.40 | <0.001* | 161 (46.8) | 19.97 | <0.001* | 438 (59.0) | 59.51 | <0.001* |
| **Sex** | | | | | | | | | |
| **Male** | 125 (68.7) | | | 72 (41.1) | | | 197 (55.2) | | |
| **Female** | 237 (58.4) | 5.64 | 0.02* | 160 (38.4) | 0.40 | 0.53 | 397 (48.2) | 4.80 | 0.28 |
| **Marital Status** | | | | | | | | | |
| **Currently Married** | 158 (56.2) | | | 81 (34.6) | | | 239 (46.4) | | |
| **Currently Unmarried** | 204 (66.5) | 6.48 | 0.01* | 151 (42.2) | 3.40 | 0.07 | 355 (53.4) | 5.65 | 0.02* |
| **Formal Education** | | | | | | | | | |
| **Yes** | 53 (47.3) | | | 108 (36.2) | | | 161 (39.3) | | |
| **No** | 309 (64.9) | 11.86 | <0.01* | 124 (42.2) | 2.19 | 0.14 | 433 (56.2) | 30.80 | <0.001* |
| **Currently working** | | | | | | | | | |
| **Yes** | 185 (56.9) | | | 90 (34.7) | | | 275 (47.1) | | |
| **No** | 177 (67.3) | 6.62 | 0.01* | 142 (42.6) | 3.81 | 0.05* | 319 (53.5) | 4.89 | 0.03* |

*Significant at p< 0.05

## Qualitative findings

**Preferred care setting.** *Ageing in place (Home).* The FGDs in both settings revealed that most of the participants preferred to remain in their homes which were either self-owned or rented. They insisted on their choice even if they had access to better structures or facilities. However, for many, the next best option was to move to their family homes which are usually intergenerational and acceptable in the traditional setting.

**Table 3. Other factors associated with the desire to age in place.**

| Location | Rural = 588 | | | Urban = 592 | | | Total = 1180 | | |
|---|---|---|---|---|---|---|---|---|---|
| Variables | Desired AIP | $\chi^2$ | p-value | Desired AIP | $\chi^2$ | p-value | Desired AIP | $\chi^2$ | p-value |
| **Home Ownership** | | | | | | | | | |
| **Owner-occupied** | 271 (68.8) | | | 197 (54.0) | | | 468 (61.7) | | |
| **Rented** | 91 (46.9) | 26.29 | <0.001* | 35 (15.4) | 87.30 | <0.001* | 126 (29.9) | 109.07 | <0.001* |
| **Living arrangement** | | | | | | | | | |
| **Living with others** | 294 (62.3) | | | 167 (38.8) | | | 461 (51.1) | | |
| **Living Alone** | 68 (58.6) | 0.53 | 0.47 | 65 (40.4) | 0.13 | 0.72 | 133 (48.0) | 0.78 | 0.38 |
| **Home Assistance** | | | | | | | | | |
| **Yes** | 301 (64.2) | | | 173 (39.1) | | | 474 (52.0) | | |
| **No** | 61 (51.3) | 6.70 | 0.01* | 59 (39.3) | 0.00 | 0.97 | 120 (44.6) | 4.58 | 0.03* |
| **Self–Rated Health (SRH)** | | | | | | | | | |
| **Good SRH** | 269 (63.3) | | | 171 (41.0) | | | 440 (52.3) | | |
| **Poor SRH** | 93 (57.1) | 1.94 | 0.16 | 61 (34.9) | 1.96 | 0.16 | 154 (45.6) | 4.32 | 0.04* |
| **Self–Rated Quality of Life (SRQOL)** | | | | | | | | | |
| **Good SRQOL** | 331 (65.4) | | | 203 (41.3) | | | 534 (53.6) | | |
| **Poor SRQOL** | 31 (37.8) | 22.73 | <0.001* | 29 (28.7) | 5.6 | 0.02* | 60 (32.8) | 26.70 | <0.001* |

*Significant at p< 0.05

**Table 4. The most significant factors associated with the preference to age in place by location.**

| Location | Rural = 588 | | | Urban = 592 | | | Total = 1180 | | |
|---|---|---|---|---|---|---|---|---|---|
| | OR | 95% CI | p-value | OR | 95% CI | p-value | OR | 95% CI | p-value |
| **Age Category (years)** | | | | | | | | | |
| ≥70 years | 2.07 | 1.37–3.14 | <0.001* | 1.54 | 1.03–2.31 | 0.04* | 1.88 | 1.42–2.49 | <0.001* |
| <70 years | 1 (ref) | | | 1 (ref) | | | 1 (ref) | | |
| **Sex** | | | | | | | | | |
| **Male** | 2.41 | 1.53–3.81 | <0.001* | 1.14 | 0.70–1.88 | 0.60 | 1.87 | 1.35–2.59 | <0.001* |
| **Female** | 1 (ref) | | | 1 (ref) | | | 1 (ref) | | |
| **Marital Status** | | | | | | | | | |
| **Currently married** | 0.61 | 0.40–0.92 | 0.02* | 0.94 | 0.59–1.50 | 0.79 | 0.80 | 0.60–1.08 | 0.15 |
| **Currently unmarried** | 1 (ref) | | | 1 (ref) | | | 1 (ref) | | |
| **Formal Education** | | | | | | | | | |
| **Yes** | 0.49 | 0.30–0.81 | 0.01* | 0.87 | 0.57–1.31 | 0.50 | 0.50 | 0.37–0.66 | <0.001* |
| **No** | 1 (ref) | | | 1 (ref) | | | 1 (ref) | | |
| **Currently working** | | | | | | | | | |
| **Yes** | 0.75 | 0.50–1.11 | 0.15 | 0.80 | 0.54–1.17 | 0.25 | 0.86 | 0.66–1.13 | 0.29 |
| **No** | 1 (ref) | | | 1 (ref) | | | 1 (ref) | | |
| **Home assistance** | | | | | | | | | |
| **Yes** | 1.79 | 1.15–2.79 | 0.01* | 0.94 | 0.62–1.43 | 0.78 | 1.32 | 0.98–1.78 | 0.07 |
| **No** | 1 (ref) | | | 1 (ref) | | | 1 (ref) | | |
| **Home-ownership** | | | | | | | | | |
| **Owner-occupied** | 2.47 | 1.69–3.61 | <0.001* | 5.83 | 3.82–8.91)* | <0.001* | 3.42 | 2.62–4.46* | <0.001* |
| **Rented** | 1 (ref) | | | 1 (ref) | | | 1 (ref) | | |
| **Self-Rated Health** | | | | | | | | | |
| **Good** | 1.44 | 0.95–2.17 | 0.08 | 1.34 | 0.89–2.01 | 0.16 | 1.40 | 1.06–1.86 | 0.02* |
| **Poor** | 1 (ref) | | | 1 (ref) | | | 1 (ref) | | |

*Significant at p< 0.

** SRQoL excluded due to colinearity

Several participants were hesitant to move in with their children, which is considered the next step in the traditional context of the provision of care with the onset of functional decline. The participants, however, did not mind having the children come around to provide assistance and help them with their activities of daily living. Women were, however, more ambivalent. Although there was a specific mention of staying in their homes with their spouses for most female participants, they were not opposed to moving to houses built by their children.

Participants were further queried to see if the preference of place to AIP would differ if there were health issues and functional decline. Mostly, participants still preferred to AIP, preferably in their homes with family members on hand to provide the necessary assistance. Although women chose to AIP, they were also receptive to being taken to their children's place in the event of functional decline or ill health. Interestingly, men also agreed to leave their homes in the event of functional decline, with the advanced directive that their family would return them to their homes once they had died.

For the older urban females, the option of staying with their sons was not an enthusiastic one as they were careful not to constitute a disturbance to their daughters-in-law. At the same time, they felt their daughters had enough responsibility to take care of their own families. Likewise, participants in the rural discussions were also receptive to moving to their

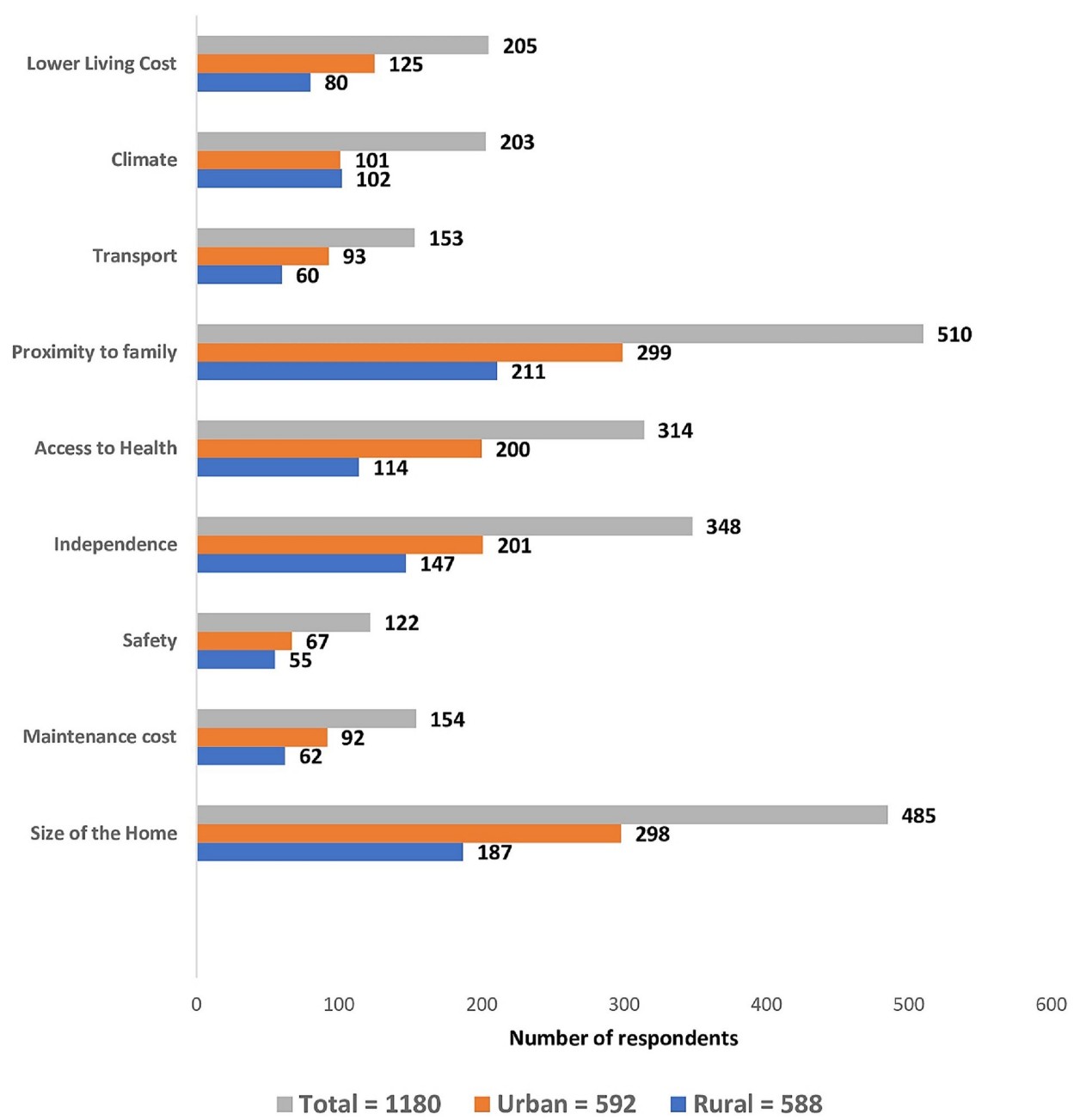

**Fig 1. Respondents' reasons for considering relocation by location.**

children's homes, especially if there is a high level of dependence. An observed disadvantage of staying with the children identified by the participants was the increasing problem of isolation.

Apart from the immediate family, participants were open to living with other family members, such as their siblings, cousins and other extended family members. Furthermore, participants were open to care by employed formal caregivers, especially if the children needed to be at work.

*Ageing in place (Community)*. Participants mentioned a preference to live within communities where such an individual has ties, friends, and relationships. Rural-urban migration was noted as a problem. According to rural participants, the best arrangement was to stay in an environment where they had peer interaction or family nearby. Also, such an individual could give back to the community.

### Determinants of the preferred setting for care

**Home ownership.** Home ownership was a determinant of the choice of place to AIP. Either owned or communal, both rural and urban participants felt that it would be more befitting to own the house. However, some homeowners indicated they would not mind relocating to another place with better facilities. Activities that give older persons a sense of purpose, such as rearing pets and greeting everybody in the community, were also considerations when deciding where to age. As such, participants emphasised that relocation may deny older adults the joy of routine they had developed over the years.

**Views about institutional care.** Participants were ambivalent about institutional care. Some participants mentioned that they don't mind staying in an older people's home built by the government if certain conditions are met. The conditions include quality of care, protection of their independence, autonomy and basic necessities of life. Even the young-old participants (<70 years) in the urban area favoured institutional care as they compared available facilities they had heard about in HIC. Likewise, the female participants were not opposed to the idea of the government providing care for older persons and setting up a facility dedicated to this purpose. Some participants also referred to private facilities run by religious bodies, which made the arrangement even more appealing.

Regarding the benefits of institutional care, participants noted that the arrangement would relieve family members of their caregiving responsibilities and therefore be an added advantage. Nevertheless, participants insisted that institutional care was acceptable so long as there was no infringement on the fundamental human right of freedom.

Participants who rejected the idea of institutional care did so on the premise that it was based on a foreign culture. Additionally, there is a high expectancy of reciprocity in the African setting. As such, the expectation is that if the older person has laboured to train the child, it was expected the child would, in turn, do the same. Also, some participants who did not support institutional care opposed the arrangement because of the cost. A common view among the participants in both settings was that the provision of care of an ageing parent ought not to be paid for since their children can provide it. Likewise, participants were fearful that the care received would most likely be sub-standard compared with available facilities in HIC. Other participants reflected that institutional care would invade the right to freedom of an individual.

### Discussion

This study determined factors associated with the preferred care setting among community-dwelling older persons in a rural and urban setting in Oyo State, south-western Nigeria and explored their views about their choices. The quantitative aspect of the study revealed that older persons in both settings prefer to AIP. Factors associated with the decision include rural setting, older age ($\geq$ 70 years), being male, home-ownership and having assistance at home. The FGD corroborated the choice of community-dwelling older persons to AIP within the home or community mostly based on the traditional expectations for reciprocity of care. Factors associated with this preference include functional status, social connectedness, and community participation. Additionally, participants were ambivalent regarding institutional care.

This study corroborates findings from other studies across income settings showing that older persons mainly desire to AIP [21, 49–51]. A possible reason for this finding may be the expected reciprocity of care in traditional settings in Nigeria and the high expectation that the family is responsible for providing LTC [38, 52, 53]. As reflected in the FGD, there is a high expectation that children will care for their parents when they grow old. However, the ongoing societal changes, such as the high level of rural-urban migration and increased female participation in the workforce, threaten this arrangement's viability. According to the focus group discussants, the best alternative was to explore other arrangements that allowed older persons to stay in an environment which facilitated peer and family interaction.

The decision to AIP may result from a strong attachment of older persons to their homes and neighbourhood in such environments [19, 49, 54]. In this study, the most frequently reported reason influencing the respondent's desire to relocate in both communities was a desire to be close to family members. This finding may be because of the younger population's high rate of rural-urban migration in search of better economic opportunities [50]. Similar to previous research, another reason the participants gave for considering relocation was the need to remain independent which may include relocating to a smaller apartment [49]. Additionally, the participants in this study frequently reported the desire for better access to health services. This finding may result from the functional and health challenges experienced with the ageing process [55]. Nevertheless, as documented in this study, other less frequently reported reasons for the desire to relocate include concerns for safety, house maintenance costs and transport difficulties.

The study revealed that more rural participants preferred to AIP in their homes or communities than urban participants. This finding is expected as many individuals have strong ties with their villages [6, 56, 57]. Also, a higher proportion of rural dwellers had always lived in the same community compared to urban dwellers. In many instances, after migrating to the urban area to work and spend their productive years, many older persons return to their villages, where they prefer to die in the hands of their loved ones. Also, if they die outside their homes, they often leave advanced directives, which likely stipulate that their remains should be buried in their towns of origin [58, 59]. Another explanation for the finding may be the increased social participation and intergenerational living arrangement in the rural area compared with the urban setting. In the FGD, participants discussed how neighbours filled the care void when family members could not offer care and support. Participants also spoke about their involvement in many community activities, including constructing boreholes and church activities. Also, some mentioned that they also cared for their grandchildren as well.

Among both the rural and urban participants, factors shown to predict a desire to AIP were older age (≥70 years) and living in owned housing, similar to research by Lofqvist and colleagues [49]. This finding may be due to the decline in health due to old age. The individual's health has been copiously documented to influence the decision to relocate or AIP [60]. Apart from familiarity within the space, home-ownership facilitates necessary structural modifications to ensure person-environment fit [61]. However, factors peculiar to the rural participants and associated with a desire to AIP were the male gender and having assistance at home. The sex difference is expected as older men are often better off financially than older females. This finding may be because often, men may have had better educational and employment opportunities. Again, older females, particularly widows, have been documented to be more financially vulnerable [3]. Also, many participants preferred to AIP mostly in their homes with family members on hand to provide the necessary assistance, care and support. The FGD corroborates that care within family homes is common in many African communities.

There were gender differences with preference of setting to AIP. The FGD revealed that many women preferred to AIP in their husband's homes. However, participants accepted that

the place to stay as they aged was flexible and could change. This acceptance was amplified if there were health issues and the individuals could not care for themselves and required a lot of assistance. Older females were more receptive to being taken to their children's place in the event of functional decline or ill health. However, a frequently mentioned problem was the possibility of isolation.

Furthermore, older women in the study were more open to care by employed formal care-givers, especially if the children needed to be away at work. However, many desired to return to their marital homes after their death. Likewise, as reflected in the FGD, older females were hesitant to stay with their sons as they were careful not to disturb their daughters-in-law. This finding is not unusual, as daughters-in-law have been previously documented as perpetrators of abuse of older women [62]. Also, educated rural participants were less likely to desire to AIP, which contradicts another research [63]. A possible reason may be that poverty levels are high in the typical rural setting and further complicated by a poor level of development and infrastructure. Both male and female participants noted a preference to live within communities where they had social ties, friends, and relationships.

However, it is surprising that many participants were not opposed to the notion of care beyond the family. Participants in this study also recognised that institutional care might be considered when there are extreme levels of disability. This change in attitude may be due to the country's high poverty level. As such, participants were more open to any form of care irrespective of their deeply held traditions and norms so long as this option would provide for all their basic needs. Additionally, as emphasised by the older persons in the qualitative research, such arrangements were acceptable so long as their fundamental rights of freedom and dignity were not infringed upon.

This finding is contrary to previously documented research and anecdotal evidence. Van der Geest reported similar findings to this study in Ghana [53]. This finding may reflect the reality on the ground whereby older adults have seen the reduced viability of sole reliance on the family [53]. The added attraction, reasoned by the participants during the FGD, is that being in an institutional setting may provide an increasingly rare opportunity for older dwellers to interact with their peers. This study's younger participants (<70 years) favoured institutional care. This finding may be because they were more likely to be educated and exposed to western culture. However, participants were more favourably disposed to institutional care if the facility was privately owned or managed by religious bodies. This finding may be because of the high level of religiosity in the country, and the trust older persons have for their religious leaders.

Participants who rejected institutionalisation as an option for LTC of older persons did so because they opined that it was based on a foreign culture. Additionally, there is a high expectancy of reciprocity in the African setting. As such, the expectation is that if the older person has laboured to train the child, the child would, in turn, do the same [38, 52, 57, 64]. Also, some participants who did not support institutional care opposed the arrangement because of the cost. As reflected in the FGD, participants believed that caring for older persons is expected to be by the children or family and not outsourced or paid for. Lehnert and colleagues, in their review, reported that married older persons mostly prefer informal LTC arrangements [65]. Also, older persons with higher incomes or who cared for an older relative were more likely to choose informal arrangements for their LTC. However, the review revealed that age, gender, and education yielded inconsistent effects [65].

Findings from the FGD revealed that participants in both locations accepted institution-based care arrangements, mainly if the facilities are provided by the government and quality is assured. There was, however, some scepticism as participants felt the government was not inclined to do such.

## Limitations of the study

This cross-sectional nature prevents inference about the causal relationship among the factors associated with the choice of care setting. Also, the study is conducted in south-western Nigeria, limiting its generalizability to older persons in other geopolitical zones with different norms and customs. Future research needs to be conducted in other geopolitical zones. The strengths however lie in the utilisation of both quantitative and qualitative measures and adds to the evidence base for policy and targeted action.

## Implications for research and clinical practice

The major implication of this study is that both the rural and urban older persons desired AIP in their homes and community. The voluntariness of this option suggests that they are not stuck in place. This choice calls for increased research into community-based provisions for older persons. Future work should consider large-scale studies on the care provision of older persons within the homes and community. Similarly, home ownership affordability and availability, including the quality of care or assistance rendered by the family, need to be studied. The increasing shift in non-resistance to institutional care among older persons in our setting needs further exploration to understand the context, structure and belief systems in the chosen setting. The information will guide the development of a non-existent model for sustainable care within the community in resource-limited settings.

## Conclusion

To our knowledge, this is the first study that explores the preferences for LTC of older persons in Nigeria. Family-based care is the predominant care system for older persons in many SSA, including Nigeria. However, the present form of family-based care cannot deliver care at the recommended quality of LTC. This study has shown that ageing in place was the choice of community-dwelling older persons in the selected communities, not because they lacked any other affordable alternative. The participants prefer to age in place even with noted functional decline. This study contributes to ongoing global efforts to improve the welfare of older persons, particularly in LMICs. Information obtained will assist in targeted action for providing community-based LTC for older persons in Nigeria and similar resource settings.

## Acknowledgments

My gratitude goes to all my research assistants, ably coordinated by Miss Odunayo Ife Ajayi, for their efforts during data collection and my wonderful participants in Igboora and Yemetu Alaadorin.

## Author Contributions

**Conceptualization:** Eniola Olubukola Cadmus, Lawrence Adekunle Adebusoye, Eme Theodora Owoaje.

**Data curation:** Eniola Olubukola Cadmus.

**Formal analysis:** Eniola Olubukola Cadmus.

**Funding acquisition:** Eniola Olubukola Cadmus.

**Investigation:** Eniola Olubukola Cadmus.

**Methodology:** Eniola Olubukola Cadmus, Lawrence Adekunle Adebusoye, Eme Theodora Owoaje.

**Project administration:** Eniola Olubukola Cadmus.

**Supervision:** Lawrence Adekunle Adebusoye, Eme Theodora Owoaje.

**Writing – original draft:** Eniola Olubukola Cadmus.

**Writing – review & editing:** Eniola Olubukola Cadmus, Lawrence Adekunle Adebusoye, Eme Theodora Owoaje.

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
