## [Decision Letter · Decision Letter 0]

6 Oct 2022

PONE-D-22-24201Ageing in place or stuck in place: Long-term care preferences of community-dwelling older persons in a low resource country in Sub Saharan Africa.PLOS ONE

Dear Dr. Cadmus,

Thank you for submitting your manuscript to PLOS ONE. After careful consideration, we feel that it has merit but does not fully meet PLOS ONE’s publication criteria as it currently stands. Therefore, we invite you to submit a revised version of the manuscript that addresses the points raised during the review process.

Please review this paper considering the reviewers' comments and orientations.

The Materials and Methods section requires significant revisions. Considering the relevance of this section for the scientific quality of the paper, our decision is for major revision. More attention is also needed to the presentation of the results, especially of the qualitative study.

Make sure that in revising the paper you consider the reviewers' indications, ensuring greater clarity, detail and accuracy on how the study was performed (method). We hope that the reviewers' comments, suggestions and orientations will help you to improve the quality of the manuscript so that it fits the PLOS ONE publication requirement.

We look forward to receiving your revised manuscript.

Kind regards,

Carla Maria Gomes Marques de Faria, Ph.D.

Academic Editor

PLOS ONE

Journal Requirements:

"This research was supported by the Consortium for Advanced Research Training in Africa (CARTA).CARTA is jointly led by the African Population and Health Research Center and the University of the Witwatersrand and funded by the Carnegie Corporation of New York (Grant No. G-19-57145), Sida (Grant No:54100113), Uppsala Monitoring Center, Norwegian Agency for Development Cooperation (Norad), and by the Wellcome Trust [reference no. 107768/Z/15/Z] and the UK Foreign, Commonwealth & Development Office, with support from the Developing Excellence in Leadership, Training and Science in Africa (DELTAS Africa) programme. The statements made and views expressed are solely the responsibility of the Fellow. For the purpose of open access, the author has applied a CC BY public copyright licence to any Author Accepted Manuscript version arising from this submission.”

"This research was supported by the Consortium for Advanced Research Training in Africa (CARTA).CARTA is jointly led by the African Population and Health Research Center and the University of the Witwatersrand and funded by the Carnegie Corporation of New York (Grant No. G-19-57145), Sida (Grant No:54100113), Uppsala Monitoring Center, Norwegian Agency for Development Cooperation (Norad), and by the Wellcome Trust [reference no. 107768/Z/15/Z] and the UK Foreign, Commonwealth & Development Office, with support from the Developing Excellence in Leadership, Training and Science in Africa (DELTAS Africa) programme. The statements made and views expressed are solely the responsibility of the Fellow. For the purpose of open access, the author has applied a CC BY public copyright licence to any Author Accepted Manuscript version arising from this submission.”

The funders had no role in study design, data collection and analysis, decision to publish, or preparation of the manuscript"

Reviewers' comments:

Reviewer's Responses to Questions

**Comments to the Author**

1. Is the manuscript technically sound, and do the data support the conclusions?

Reviewer #1: Yes

Reviewer #2: Partly

2. Has the statistical analysis been performed appropriately and rigorously? 

Reviewer #1: Yes

Reviewer #2: Yes

3. Have the authors made all data underlying the findings in their manuscript fully available?

Reviewer #1: Yes

Reviewer #2: Yes

4. Is the manuscript presented in an intelligible fashion and written in standard English?

Reviewer #1: Yes

Reviewer #2: Yes

5. Review Comments to the Author

Reviewer #1: I enjoyed reading this article, due to the relevance and topicality of the theme.

At the present time, when aging in place is defended so much, this study raises a very pertinent question, which is to what extent the AIP is a choice of the aging person or a forced choice due to lack of alternatives.

This study adds something new as it focuses on the LMIC, where there is a lack of studies that analyze this issue. In fact, most studies are carried out in HIC, where there is a clear preference for AIP, largely due to the resources of this population. This study seems to me also relevant because of the novelty of the topic, given that, as the authors say, it is the first study to look for AIP determinants in Nigeria.

Overall, this paper is likely to be of interest to the PLOS ONE readership, but I have some suggestions for improvements to the manuscript.

Abstract

- The objective presented refers to the qualitative study and the method only refers to what was done at a qualitative level. Here it would be important to give a broader view of the study, including the quantitative study, because in this way the idea of the mixed study is lost.

- As for the results, I suggest that you present the categories and sub-categories within each theme and write one or two sentences that give an overview of the results.

Introduction

- The introduction presents a good demographic framework on aging in Africa and specifically in Nigeria, contextualises the concern with AIP options in the context of LMIC and introduces relevant concepts for understanding the topic.

- The expression “older populace”, which appears for the first time in line 62 and then is repeated throughout the article, should be replaced by the expression older/elderly population.

- The abbreviation QoL appears for the first time on line 80, so it must be spelled out in full.

- In lines 87 and 88, the references are written in full instead of being numbered. The referencing must be consistent throughout the article. This aspect should be reviewed.

- The Introduction ends with the objective of the study, but it could be clearer. I suggest that objectives be formulated for qualitative and quantitative studies and that the articulation between the two be clarified.

Materials and methods

- It would be important to start this section with the justification for choosing the mixed methodology. I suggest that the articulation between the data from the quantitative and qualitative studies be clearly explained and also clearly explained what the second study (which it was also necessary to say) added to the first.

- This section mentions that the methodology has already been described previously in other studies that are referenced, but I suggest that the methodology is also presented here, albeit briefly. This so that the reader does not have to resort to other articles to understand what was done in this one.

- Methodological procedures should be clearer and more complete, separating and clearly presenting what was done in the quantitative and qualitative study – for example, how many participants in each study, what are the inclusion criteria in each study, instruments used in both studies. .

- What does the acronym LGA, which appears on line 174, mean?

- It is very important to describe how the qualitative data were analyzed – What is the procedure, what is the unit of analysis, etc.

- Regarding the quantitative study, the variables analyzed and the instruments used to evaluate them must be presented.

Results

- In general, the quantitative results are well presented. The tables are clear and the description is clear.

- After line 247 (there is no longer any reference to the lines) it says: “Other factors associated with a desire to AIP among the rural participants were having owned homes, assistance at home and good self-rated quality of life (SRQoL)”. – How was quality of life assessed in this study? It is important to clarify this in the methodology part.

- In general, the qualitative results are quite complete and in-depth. I appreciate the effort to have made this presentation so exhaustive. However, I suggest that the presentation of the results be reduced to representative data, so that the essence of the results is not lost. I suggest that this section be reorganized, in order to make the results clearer and more comprehensive and to make reading more fluid and easier.

- The themes, categories and sub-categories that resulted from the qualitative analysis should be presented explicitly in the text, along with their meaning and more representative examples.

- The transcripts to illustrate the categories and sub-categories should be shorter and the most representative. When there are too many transcripts, the reader gets lost in the text and misses the essence of the results.

Discussion

The discussion articulates well the data from qualitative and quantitative studies and the results with evidence from other studies.

I suggest that after the objectives have been reformulated in the introduction, they are addressed in the discussion, showing the extent to which this study has allowed to meet them.

Reviewer #2: To the authors,

The paper “Ageing in place or stuck in place: Long-term care preferences of community-dwelling older persons in a low resource country in Sub Saharan Africa”, seems to me to be very relevant, as research on ageing is predominant in Euro-American countries.

However, in terms of content it is important to distinguish "Aging in place" from "Long-term care" in the introduction, considering that in a continuum of care we are practically in opposite poles. This problem is present in the whole article, from introduction, to method, to the Table titles in the Results section. Therefore, when problematizing the topic under investigation, it is necessary to precisely define the mains concepts in order to clarify study goals. The analysis of some systematic reviews in the field may help to clarify the subject (Rosenwohl-Mack et al., 2020; Ahn, 2017).

Regarding the method section, the mixed-methods design is particularly interesting. Simply, because it is a demanding methodology, it is essential to present in sufficient detail each phase of the study (first Quantitative, followed by Qualitative).

Quantitative - the sample and sampling are very clear. On the other hand, data collection instruments and procedures are practically omitted. For example, there is reference to a Questionnaire. What kind of questionnaire? With open questions? Closed Questions? How many items? On what topics? The tables in the results section provide some information regarding questions/items, but the information available to readers is insufficient. For example, in the logistic regression, it is necessary to indicate which variables are constants of the regression model and inclusion criteria, if applicable.

Qualitative - the use of Focus Group is very interesting. There is some information, but important details are missing. For example, how were the task distributed within the team? Who led the focus group? Who was responsible for the dynamics of the group? How was the problem of simultaneous speech solved both in the management of the group and later in the transcription?

As for the results, the sequence of presentation seems to be clear: (1st) quantitative data and (2) qualitative data. However, there are some aspects that need to be improved.

Quantitative data: Verify tables designations in view of the objectives of the study, i.e., if “aging in place” or "long-term care". Qualitative data: in quotations from the focus group, besides age group, authors should add the participants’ age. If we have more information about the participants (age, sex, area of residence, etc.) we can more easily observe trends in the variables under study and especially perform a more dynamic interpretation of quantitative and qualitative results.

Discussion and conclusion: it is important to highlight the main findings (quantitative and qualitative) in order to discuss research trends. At the same time, the authors must clarify if the paper main contributions address "aging in place", "long-term care" or both. It is necessary to include study limitations.

Other aspects to consider:

The article is written very clearly. However, there are some imprecisions, as the number of participants in the method section – Sampling Technique, line 183 [Similar to previous studies, if the 58 [588?] selected respondent….].

Equally, the in-text citations do not always follow the same reference norms throughout the paper. Even though superscript numerals predominate, there are cases with publication year. It is recommended to correct. Example Introduction, line 96 [For instance, Greenfield (2016=, in a qualitative study among older persons….]

6. PLOS authors have the option to publish the peer review history of their article (what does this mean?). If published, this will include your full peer review and any attached files.

Reviewer #1: **Yes: **Diana Maria da Costa Bizarro Morais

Reviewer #2: No

---

## [Author Response · Author response to Decision Letter 0]

27 Mar 2023

Dear Sir,

Thank you for the detailed review of my manuscript. Please find below the response to the comments made.

Thank you.

E. Cadmus

Serial Number Comment Response

 Reviewer 1 

1. Abstract

- The objective presented refers to the qualitative study and the method only refers to what was done at a qualitative level. Here it would be important to give a broader view of the study, including the quantitative study, because in this way the idea of the mixed study is lost. ‘This study determined factors associated with the preferred care setting among community-dwelling older persons (quantitative objective) and explored their views about their choices (qualitative objective)

 Line 18-19.

 - As for the results, I suggest that you present the categories and sub-categories within each theme and write one or two sentences that give an overview of the results. Done (Table 4)

 Introduction 

 The expression “older populace”, which appears for the first time in line 62 and then is repeated throughout the article, should be replaced by the expression older/elderly population Revised (Line 58)

 The abbreviation QoL appears for the first time on line 80, so it must be spelled out in full.

 Revised (Line 82)

 In lines 87 and 88, the references are written in full instead of being numbered. The referencing must be consistent throughout Revised (Line 89 and 91)

 Introduction 

 - The Introduction ends with the objective of the study, but it could be clearer. I suggest that objectives be formulated for qualitative and quantitative studies and that the articulation between the two be clarified. This study compared using quantitative measures the preferred care setting for ageing and factors associated with the decision among rural and urban community-dwelling older persons aged 60 years and above in Oyo State, Nigeria. Also, the study explored using qualitative measures the views about their choices

Revised ( Line140-143)

 Materials and methods 

 It would be important to start this section with the justification for choosing the mixed methodology. I suggest that the articulation between the data from the quantitative and qualitative studies be clearly explained and also clearly explained what the second study (which it was also necessary to say) added to the first.

 Revised ( Line 146-157)

 This section mentions that the methodology has already been described previously in other studies that are referenced, but I suggest that the methodology is also presented here, albeit briefly. This so that the reader does not have to resort to other articles to understand what was done in this one. Revised ( Line 158)

 Methodological procedures should be clearer and more complete, separating and clearly presenting what was done in the quantitative and qualitative study – for example, how many participants in each study, what are the inclusion criteria in each study, instruments used in both studies. Revised

Number of participants in the quantitative survey= 1180

22 FGD Total participants 172 older males and females)

Inclusion and exclusion criteria- (Line (225-234)

Adults aged 60 years and above and permanently living continuously in the selected communities for at least 12 months preceding the time of the study were eligible to participate in the study. The eligibility was based on the criteria used in a similar community-based survey conducted by Gomez-Olive et al. (2018) in South Africa [45]. However, older persons who had any form of incapacitation or health conditions such as dementia, speech or hearing disorders and other communication problems and could not personally provide information were excluded from the study. Proxies were not utilized for this study because the objective was to document the older person’s views about their preferences and ageing experience, which a proxy may not accurately capture.

 What does the acronym LGA, which appears on line 174, mean? Local Government Area (LGA) Fully Spelt line 165

 It is very important to describe how the qualitative data were analyzed – What is the procedure, what is the unit of analysis, etc.

 Revised Line 267-282-

 Materials and methods 

 Regarding the quantitative study, the variables analyzed and the instruments used to evaluate them must be presented Line 298-324

 Results 

 In general, the quantitative results are well presented. The tables are clear, and the description is clear. Noted

 After line 247 (there is no longer any reference to the lines) it says: “Other factors associated with a desire to AIP among the rural participants were having owned homes, assistance at home and good self-rated quality of life (SRQoL)”. – How was quality of life assessed in this study? It is important to clarify this in the methodology part. Participants attitude to ageing was assessed using the attitude to ageing questionnaire (AAQ). This measured the attitude in psychosocial, physical change and psychological domains. Participants were asked to rate their quality of life on a five-point Likert scale ranging from very satisfied to very bad. The responses were categorized as having good QoL (if their response to the question of Qol were “good” or “very good” with their Qol. Likewise, respondents were said to have poor QoL if they reported these to be “bad”, “very bad”, or “neither bad nor good”; with their QoL. Participants were asked to rate their health on a five-point Likert scale ranging from very good to very bad. The responses were categorized as having good self-rated health SRH (if their response were “good” or “very good”. Likewise, respondents were said to have poor SRH if they reported these to be “bad”, “very bad”, or “neither bad nor good”; The respondent’s level of functioning was measured using the 12- item World Health Organization Disability Assessment

 In general, the qualitative results are quite complete and in-depth. I appreciate the effort to have made this presentation so exhaustive. However, I suggest that the presentation of the results be reduced to representative data, so that the essence of the results is not lost. I suggest that this section be reorganized, in order to make the results clearer and more comprehensive and to make reading more fluid and easier.

- The themes, categories and sub-categories that resulted from the qualitative analysis should be presented explicitly in the text, along with their meaning and more representative examples. Done (Table 4)

 - The transcripts to illustrate the categories and sub-categories should be shorter and the most representative. When there are too many transcripts, the reader gets lost in the text and misses the essence of the results. Done (Table 4)

 Discussion 

 The discussion articulates well the data from qualitative and quantitative studies and the results with evidence from other studies.

I suggest that after the objectives have been reformulated in the introduction, they are addressed in the discussion, showing the extent to which this study has allowed to meet them. Revised

 Reviewer #2 

 The paper “Ageing in place or stuck in place: Long-term care preferences of community-dwelling older persons in a low resource country in Sub Saharan Africa”, seems to me to be very relevant, as research on ageing is predominant in Euro-American countries.

However, in terms of content it is important to distinguish "Aging in place" from "Long-term care" in the introduction, considering that in a continuum of care we are practically in opposite poles. This problem is present in the whole article, from introduction, to method, to the Table titles in the Results section. Therefore, when problematizing the topic under investigation, it is necessary to precisely define the mains concepts to clarify study goals. The analysis of some systematic reviews in the field may help to clarify the subject (Rosenwohl-Mack et al., 2020; Ahn, 2017). The manuscript has been revised to reflect strictly the preferred care setting for older persons. i.e Aging in place within the home or community compared to other options such as moving in with the child, relatives, or care by religious organizations or within institutional settings or other

 Methods 

 Regarding the method section, the mixed-methods design is particularly interesting. Simply, because it is a demanding methodology, it is essential to present in sufficient detail each phase of the study (first Quantitative, followed by Qualitative). Revised to reflect what was done in the qualitative and quantitative aspects.

 Quantitative - the sample and sampling are very clear. On the other hand, data collection instruments and procedures are practically omitted. For example, there is reference to a Questionnaire. What kind of questionnaire? With open questions? Closed Questions? How many items? On what topics? A semi-structured questionnaire was developed reflected the study objectives and consisted of the following sections:. 

Sociodemographic characteristics and family dynamics

Current health status and selected health risk behaviours

Attitude to ageing

Assessment of Quality of life 

Assessment of self-reported functioning and disability

 The tables in the results section provide some information regarding questions/items, but the information available to readers is insufficient. For example, in the logistic regression, it is necessary to indicate which variables are constants of the regression model and inclusion criteria, if applicable. Variables that are already proven in the literature to be related to the outcome (preference for AIP) were included in the regression model and fitted for the rural, urban, and total population at a 5% statistical significance level. SRH was excluded because of collinearity. The final model was presented. 

 Qualitative - the use of Focus Group is very interesting. There is some information, but important details are missing. For example, how were the task distributed within the team? Who led the focus group? Who was responsible for the dynamics of the group? How was the problem of simultaneous speech solved both in the management of the group and later in the transcription?

As for the results, the sequence of presentation seems to be clear: (1st) quantitative data and (2) qualitative data. However, there are some aspects that need to be improved Quantitative data: Verify tables designations in view of the objectives of the study, i.e., if “aging in place” or "long-term care". The distribution of tasks for the FGD has been spelt out. 

The facilitator of each discussion was responsible for the dynamics of the group.

Ground rules were set to avoid simultaneous speech and the facilitator ensured everyone had an opportunity to voice their opinions without interruption or overruling.

The objective of the study was the preferred setting for care. i.e ageing in place (home or community)

 Qualitative data: in quotations from the focus group, besides age group, authors should add the participants’ age. If we have more information about the participants (age, sex, area of residence, etc.) we can more easily observe trends in the variables under study and especially perform a more dynamic interpretation of quantitative and qualitative results. Age, Sex and Area of residence already included in the quotes. The groups were stratified based on Sex (Male, Female) ,age ( 60-70 and ≥70 years) and Area of residence (rural urban).

Each quote is thus labelled in that format. For example, Male FGD_ 70 years and above_ Urban

 Discussion and conclusion 

 It is important to highlight the main findings (quantitative and qualitative) in order to discuss research trends. At the same time, the authors must clarify if the paper main contributions address "aging in place", "long-term care" or both. It is necessary to include study limitations 

The main findings of the study has been summarized and included

The papers main contribution addresses Aging in Place

Study limitations included

Other aspects to consider 

 Imprecisions, as the number of participants in the method section – Sampling Technique, line 183 [Similar to previous studies, if the 58 [588?] selected respondent….].

Equally, Corrected

 in-text citations do not always follow the same reference norms throughout the paper References have been corrected

 Even though superscript numerals predominate, there are cases with publication year. It is recommended to correct. Example Introduction, line 96 [For instance, Greenfield (2016=, in a qualitative study among older persons….] Corrected

---

## [Decision Letter · Decision Letter 1]

3 Aug 2023

PONE-D-22-24201R1Ageing in place or stuck in place: Preferred care setting for community-dwelling older persons in a low-resource country in Sub Saharan Africa.PLOS ONE

Dear Dr. Cadmus,

Thank you for submitting your manuscript to PLOS ONE. After careful consideration, we feel that it has merit but does not fully meet PLOS ONE’s publication criteria as it currently stands. Therefore, we invite you to submit a revised version of the manuscript that addresses the points raised during the review process.

We look forward to receiving your revised manuscript.

Kind regards,

Carla Maria Gomes Marques de Faria, Ph.D.

Academic Editor

PLOS ONE

Journal Requirements:

**Additional Editor Comments:**

Consider the reviewer's comments and suggestions regarding the methods section, in particular regarding the sequence of presentation of the information (first, the quantitative approach (sample, data collection protocol, analytic strategy, and other information deemed convenient); only then the qualitative approach (i.e., focus group).

Reviewers' comments:

Reviewer's Responses to Questions

**Comments to the Author**

1. If the authors have adequately addressed your comments raised in a previous round of review and you feel that this manuscript is now acceptable for publication, you may indicate that here to bypass the “Comments to the Author” section, enter your conflict of interest statement in the “Confidential to Editor” section, and submit your "Accept" recommendation.

Reviewer #1: All comments have been addressed

Reviewer #2: (No Response)

2. Is the manuscript technically sound, and do the data support the conclusions?

Reviewer #1: Yes

Reviewer #2: Yes

3. Has the statistical analysis been performed appropriately and rigorously? 

Reviewer #1: Yes

Reviewer #2: Yes

4. Have the authors made all data underlying the findings in their manuscript fully available?

Reviewer #1: Yes

Reviewer #2: Yes

5. Is the manuscript presented in an intelligible fashion and written in standard English?

Reviewer #1: Yes

Reviewer #2: Yes

6. Review Comments to the Author

Reviewer #1: (No Response)

Reviewer #2: Comments to the authors,

The current version of the manuscript “Ageing in place or stuck in place: Preferred care setting for community-dwelling older persons in a low-resource country in Sub Saharan Africa” presents substantial improvements. The revised manuscript gains in clarity, interest and potential impact.

I believe the adjustments made to the article subtitle – “Preferred care setting for community-dwelling older persons in a low-resource country in Sub Saharan Africa” – are pertinent and adequate. The subtitle now better conveys what is nuclear in the study: researching older adults’ (60+ years) preferences regarding care settings in an aging in place framework at an urban and rural level.

From my perspective, using a mixed-methods design is one of this study’s strengths. However, the methods section needs further adjustments because the presentation sequence used for the research plan (qualitative approach followed by quantitative approach) is precisely the reverse of what is shown in the results section. Also, there is scattered information that needs to be associated with the respective methodological approach.

Considering that the results are presented in an intelligible sequence, I suggest that the research plan, in the methods section, is presented in the same order: (1) first, the quantitative approach (sample, data collection protocol, analytic strategy, and other information deemed convenient); (2) only then the qualitative approach (i.e., focus group). Additionally, all the information that concerns each study approach should be associated with the approach it refers to. In other words, a better organization of contents in the methods section will make the research plan more comprehensible.

Since the samples are independent, I believe that following this sequence when presenting both the research plan and the results better supports the authors’ explicitly stated decision to interpret the results as a whole in the discussion/conclusion.

In what refers to the results, it is noteworthy that the revisions made by the authors contributed to a more robust and interesting paper. Notably, the results concerning the qualitative approach (focus group) substantially improved.

Finally, I would like to highlight the relevance of the content added to the various sections of the Discussion. This study’s potential impact on population aging – in its implications for policy, practice and future research – considerably gained.

Sincerely,

Reviewer

7. PLOS authors have the option to publish the peer review history of their article (what does this mean?). If published, this will include your full peer review and any attached files.

Reviewer #1: No

Reviewer #2: No

---

## [Author Response · Author response to Decision Letter 1]

18 Sep 2023

Dear Sir,

Thank you for the detailed review of my manuscript. 

Please find below the response to the comments made.

Thank you.

E. Cadmus

Reviewers Comment 

1. The methods section needs further adjustments because the presentation sequence used for the research plan (qualitative approach followed by quantitative approach) is precisely the reverse of what is shown in the results section.

Response Page Number Methods section has been edited in the suggested order

-first, the quantitative approach

-then the qualitative approach (i.e., focus group). Pages 12-18

2. There is scattered information that needs to be associated with the respective methodological approach. 

Response and Page Number

Noted scattered information has been associated with the respective methodological approach. -

3. I suggest that the research plan, in the methods section, is presented in the same order: (1) first, the quantitative approach (sample, data collection protocol, analytic strategy, and other information deemed convenient); (2) only then the qualitative approach (i.e., focus group).

Response and Page Number 

The research plan, in the methods section, has been edited and presented in the suggested order: (1) first, the quantitative approach (sample, data collection protocol, analytic strategy, and other information deemed convenient); (2) then the qualitative approach (i.e., focus group). Pages 12-18

4. All the information that concerns each study approach should be associated with the approach it refers to. In other words, a better organization of contents in the methods section will make the research plan more comprehensible 

Response Page Number

Contents in the methods have been re-organised Pages 12-18

---

## [Editor Report · Decision Letter 2]

3 Oct 2023

Ageing in place or stuck in place: Preferred care setting for community-dwelling older persons in a low-resource country in Sub Saharan Africa.

PONE-D-22-24201R2

Dear Dr. Cadmus,

We’re pleased to inform you that your manuscript has been judged scientifically suitable for publication and will be formally accepted for publication once it meets all outstanding technical requirements.

Kind regards,

Carla Maria Gomes Marques de Faria, Ph.D.

Academic Editor

PLOS ONE
---

## [Editor Report · Acceptance letter]

6 Oct 2023

PONE-D-22-24201R2 

Ageing in place or stuck in place: Preferred care setting for community-dwelling older persons in a low-resource country in Sub Saharan Africa. 

Dear Dr. Cadmus:

I'm pleased to inform you that your manuscript has been deemed suitable for publication in PLOS ONE. Congratulations! Your manuscript is now with our production department. 

Kind regards, 

on behalf of

Professor Carla Maria Gomes Marques de Faria 

Academic Editor

PLOS ONE